# Demographic Spatialization Simulation under the Active "Organic Decentralization Population" Policy

## Fang Liu *, Weilun Sun and Ge Peng

School of Geomatics and Urban Spatial Informatics, Beijing University of Civil Engineering and Architecture, Beijing 100044, China
* Correspondence: LF@bucea.edu.cn; Tel.: +86-13810725660

**Abstract:** A matter of considerable concern is managing rapid population growth in a growing megacity. After years of endeavor, the "decentralize and population cap" policy has finally been implemented and has achieved some success in Beijing, China. Before applying what has been learnt from this experience to other places, two questions remain to be addressed: "Can urbanization result in land-population harmony under the double effects of accessible guiding plans and invisible push-pull forces?" and "What will be the likely geo-simulation of population density resulting from a city decentralization process?" Under the guidance of "orderly city development" theory, our research (1) simulated the effects of the "organic population decentralization" policy on future population density dynamics; (2) proposed a new framework that coupled models of Verhulst logistic differential population and Cellular Auto-Markov; and (3) analyzed the steering effect of the policy toward a spatial population distribution that could be described as "spread through decentralization." The results showed that Beijing is currently at the beginning of the "suburbanization" stage. This study can help geographers obtain an innovative method that couples the existing spatial population patterns and the potential population size, which is beneficial for urban planners in determining the spatial structure of a relative equilibrium status for urban development.

**Keywords:** organic decentralization population policy; policy steering effect; Verhulst population model; polarization; orderly; spatial simulation; suburbanization

## 1. Introduction

Considering megacity human-land disharmonies, a migrant planning strategy is an important intervention as an alternative. According to the growth pole theory proposed by Perroux and extended by Parr, growth "appears as several growth points or poles with varying intensities, and it spreads along various channels"; this results in "the growth pole effect successfully triggering the adjustment of population density distribution between dense and loose, and the growing metropolitan areas will sprawl" [1]. However, when the spatial expansion of the population develops to a certain point, further growth of the central city will cause a surge in the cost of capital, labor, and time, which poses a bottleneck for high economic productivity at high population loads. At this point, the city will not be able to provide suitable infrastructure or a decent quality of life for its citizens. When desired living conditions cannot be met in the city, the residents will naturally spread to the surrounding areas, forming multiple suburban centers, satellite towns, or a polycentric urban region with social-function divisions. The city will then move from the "agglomeration effect" to enter the "crowding out" period [2].

The concept of "orderly development" plays an important role in city management in cities all around the world. Cities go through three typical stages during their spiral development modes, i.e., polarization, orderly, and transitional stages [3]. The spiral mode means that the cities develop according to the B—A—B—C mode. Here, the meaning of "B—A—B—C mode" is the spiral growth of a creative city, from polarization to transitional

stages, then from transitional to orderly stages, and so on. Figure 1 illustrates this. During this stage, the exchange of resources (e.g., capital and labor) between central cities and their affiliated towns reaches a harmonious status, namely, "orderly" city development [4,5]. Generally, the essence of orderly city development is closely related to the redistribution of settlement in the central and suburban areas, which helps to balance the population size with the available urban resources, assists in promoting harmonious human–environment interactions, results in a reasonable urban-rural flow ratio, optimizes the population structure, and helps with the coordinated deployment of various elements, such as reasonable urban-rural flow ratios (see Figure 1) [6].

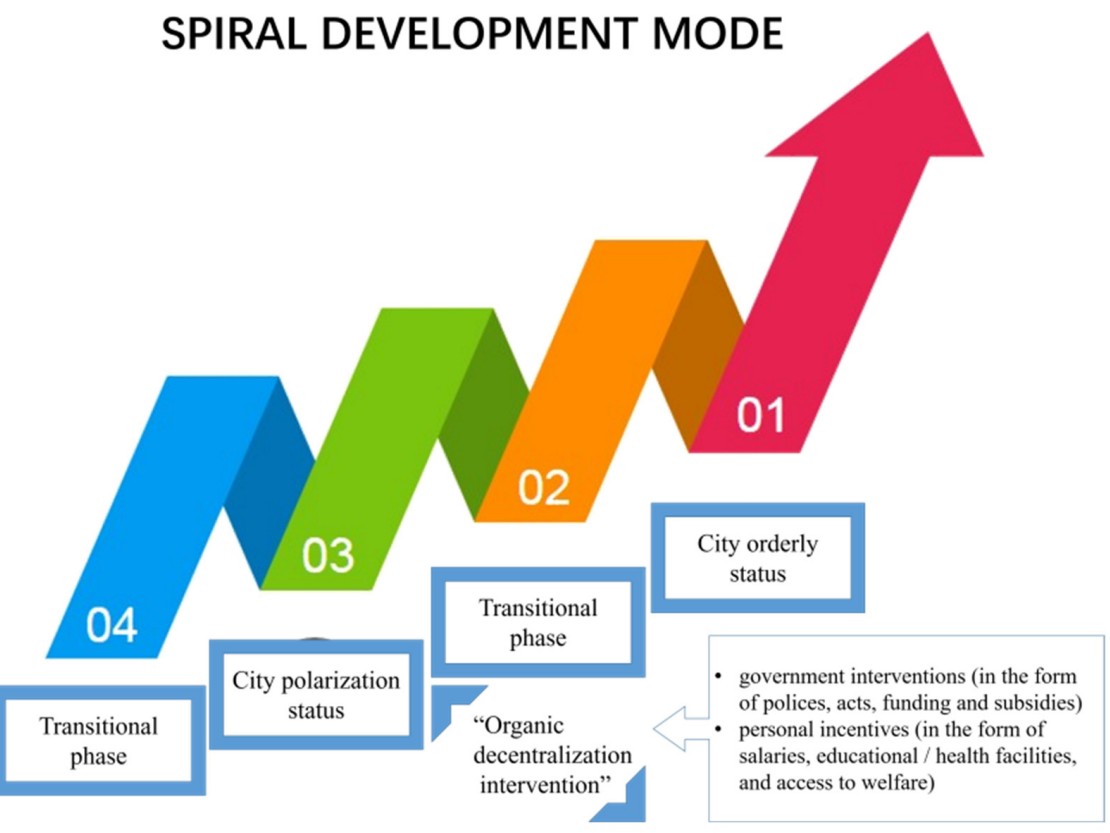

**Figure 1.** Role of "orderly development" theory and strategies during cities' spiral development modes. (For the interpretation of polarization, orderly, and transitional stages).

Population distribution is influenced by the dual effect of government interventions and personal incentives, so addressing distribution disharmony should also start from policy implications and personal choices [7]. For example, the urban super-giants of Mumbai, San Paulo, and Jakarta are examples of this urban population-economic development disassociation [8]. This phenomenon needs extra attention to repair the balance between central and suburban population distribution. History has shown that governments have adopted abundant population management policies, which often have a track record of failure [8]. Recently, an increasing number of geographers and urban planners have been promoting the decentralization of the population by reducing the multi-functionality of cities, guiding house prices, or influencing enterprise relocation. All these strategies have one aim: making the central city districts less attractive to potential immigrants. Besides policy, the invisible push-pull force also plays a role in personal choice. By improving socio-economic opportunities (education, employment, and facilities) in the outskirts of cities or in satellite cities, even inner city residents can be attracted away from the densely populated city centers to the less dense peripheries [9].

Beijing experienced the polarization development stage from 1949 to 2015, with increasing population pressure always present. The year 2016 can be regarded as a turning point and start of the inner "orderly" stage, coinciding with the release of the long-term "Beijing General Urban Planning (2016–2035)" report [10]. In this research, the citation of "organic decentralization project" is originally proposed in "Beijing General Urban Planning (2016–2035)", cited in [6]. To flatten the spatial density curve in the central six districts, prevent the further fanatical migrants influx from the suburbs and other provinces, and promote land urbanization in the suburbs, this population cap policy has been proposed through in-depth investigation. During 2016–2035, the effects of both the invisible push-pull force and the population cap policy should force more people to move outward, forming new secondary cities in the Beijing-Tianjin-Hebei region, signifying the beginning of the high-level equilibrium "city clusters" stage. Spatial characteristics gradually create integrated spatial networks by connecting urban and rural spaces [11]. Incentives and enforcement work together to both form and manage the appropriate distribution of the population to the hinterland.

There are lots of measuring indices to evaluate the performance of migration governance. The Migration Data Portal provides migration policy indices, such as the sustainable development goal (SDG) indicator 10.7.2, electoral law indicators, a global migration barometer, and migration governance indicators [12]. Overall, an increasing number of cases indicate that migration policies work through personal incentives and individual choices, and these underlying factors include employment, education, facilities and resources, and environment.

Systems theories and evaluation tools are used to guide the harmonious development of cities and their inhabitants. For simulation methods for demographic spatialization, the available techniques can be divided into four categories [13–17]:

(1) Experience-based forecasting models: these include the Gray Model (GM) and Markov Model.
(2) Quantitative prediction models: these are based on mechanical processes, such as the system dynamics (SD) model, multivariate statistical analysis method, linear programming method, and machine learning algorithms [13,14].
(3) Process simulation models: these are dynamic units, such as the cell-based cellular automata (CA) model and agent-based model (ABM) [15,16].
(4) Hybrid simulation models: these use variables to define the relative response (elasticity) of the land use type to conversion or the land use change in relation to the socio-economic and bio-physical driving factors at a small region scale, such as (CLUE-S), ABM-CA, and most global simulation models [17,18].

The essence of urbanization is population redistribution in urban and rural areas [19]. As cities experience different development stages, the size and spatial densities of the population exhibit various characteristics, e.g., population expansion during the "spread through growth" stage, or population shrinkage in the core regions during the "decentralization through growth" stage [4]. The OECD and China Development Research Foundation have reported that the spatial pattern of population distribution and the dynamics of population decentralization and centralization not only reveal the stage of city development but also the health of the city.

Although there are many methods available for predicting population distribution, few have focused on the dynamics of city inhabitant size under active "organic decentralization population" policies, which may solve "overpopulation" during urbanization processes within most megacities, as well as provide a solution to "city clusters" and "city shrinkage" [20–22]. The first challenge is that the performance of socio-economic policy is difficult to express and embed quantitatively into a simulation mechanism [23]. The second challenge of simulating inhabitant distribution originates from the dual-effect of push-pull theory and immigrant demography theory. It is difficult for the single type category of models to perform these simulations; however, the integrated models could supplement the shortcomings of a single model. The population density distribution that

matches the "orderly" development plan is unknown. This study aims to simulate and examine the performance of the "organic decentralization population" policy to determine the most sustainable spatial population densities for orderly development. In this research, an integrated constrained Verhulst and Cellular Auto-Markov model (CA-Markov) was proposed, and the modelling data comes from the statistical yearbooks and Remote Sensing (RS) image interpretation products, while validated by district-level street statistics data.

## 2. Study Area and Data Sources

This study aims to simulate the performance of the "organic decentralization population" policy to the sustainable population development. We selected Beijing as the research area for this study because it has been experiencing an active "organic decentralization population" project, which may provide a solution to "city clusters" (see Figure 2). Hence, the city will experience the results of policy-oriented population redistribution and regional re-scaling. The population in the six core regions will reduce, while the other ten suburban districts will attract migrants.

In Table 1, the permanent resident population data were derived from statistical yearbooks, while the spatial data were sourced from remote sensing image interpretation products, validated by district-level street statistics. In addition, the vector data of land urbanization was available, including those for rural residential areas, factories, mines, and other construction land.

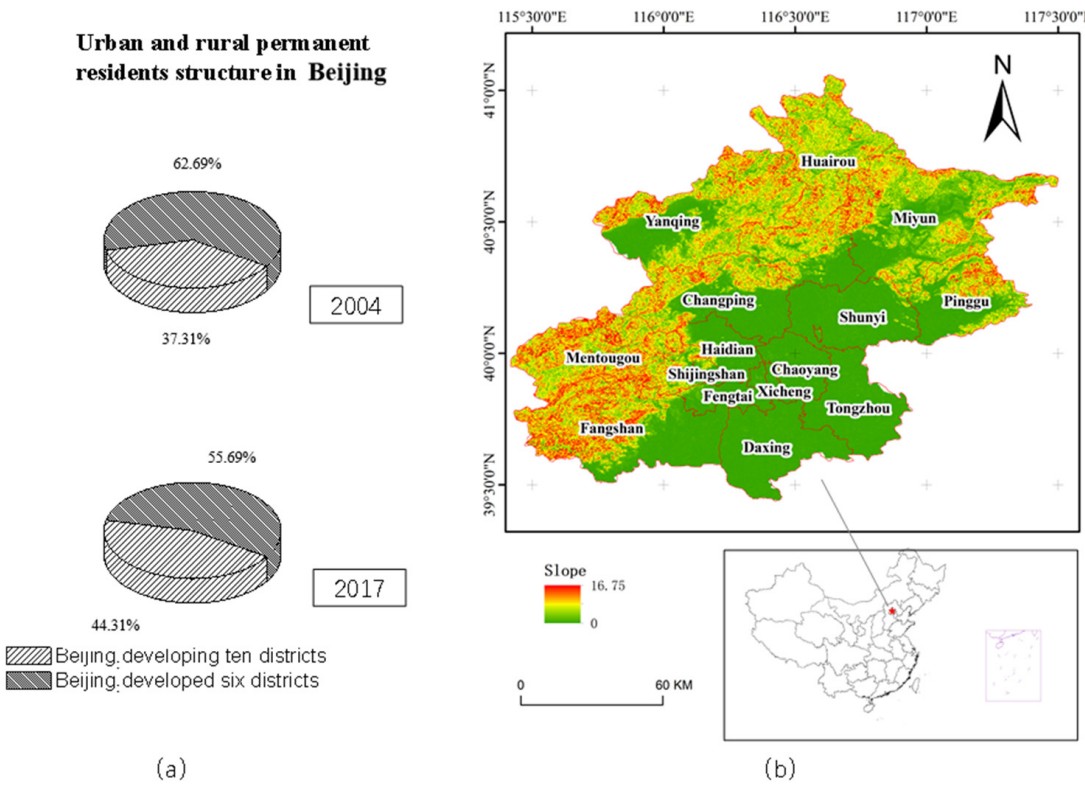

**Figure 2.** Overview of the study area and the urban and rural habitant structure of Beijing in 2004 and 2017. (**a**) The size of permanent habitants in six developed districts and ten developing districts of Beijing in 2004, and 2017 (left side of the figure); (**b**) Beijing administrative zoning map (right side of the figure).

**Table 1.** List of data sets used in the production of the population density potential map.

| | Data Sets | Date | Purpose | Sources |
|---|---|---|---|---|
| 1 | China population: Spatial Distribution Kilometer Grid Data Set (Raster file) | 2005, 2010 | CA-Markov model: for prediction | Global Change Research Data Publishing & Depository, CHN [24] |
| 2 | Electronic Map Products of Beijing (Raster file) | 2015 | | Geographical Information Monitoring Cloud Platform, CHN [25] |
| 3 | Beijing municipal data (SHP file) | 2015 | | |
| 4 | China population: Spatial Distribution Kilometer Grid Data Set (Raster file) | 2015 | for validation of the CA-Markov model | Resource and Environment Science and Data Center, CHN [26] |
| 5 | Population census data set (Statistics) | 2014–2017 | Verhulst model: for prediction of model | Beijing statistics yearbook from 2015 to 2019, CHN [27] |
| 6 | Population census data set (Statistics) | 2018 | for validation of the Verhulst model | |
| 7 | City-level and zone-level population size top-limit (policy proposals) | 2020, 2035 | for modifying model results | Beijing General Urban Planning (2016–2035) [10] |

## 3. Models and Methods

Based on the data mentioned above, the objective of this study is to explain the internal logic of the simulation of the population mobility trends after implementation of the active "organic decentralization population" policy, using the spatial dependence between the spatial population pattern and facilities. In this research, a workflow of the integrated constrained Verhulst and CA-Markov model was proposed (see Figure 3).

Three main steps were involved. First, data were prepared for the fundamental research. Second, the simulation of the spatial pattern and the simulation of the sizes in the sub-region were undertaken. Third, the new coupling model guided by the Beijing General Urban Planning report (2016–2035) was applied [10].

### 3.1. Population Density Dynamic Model under Population Cap Constraint

CA-Markov is a prediction procedure combined with the Markov Chain, logistic regression, cellular automata, and multi-objective land allocation (MOLA) modules [17]. In this study, ten different municipal facilities contributing to population density grades were chosen as driving factors, from level 1 to level 20 [28,29]. The Markov Chain module studied two historical population density grid images and exported one transition area matrix, indicating the possibility of a to-be-changed pixel, considering the likely spatial distribution transitions [30].

Before we discuss the integrated constrained CA-Markov Model with the Verhulst Logistic Model, we first illustrate the Verhulst logistic population model with maximum environmental capability (see Equations (1) and (2)) [31].

Verhulst logistic population model by François Pierre:

$$\frac{dp}{dt} = r \cdot p \cdot \left(1 - \frac{p}{p_m}\right), \tag{1}$$

$$p = \frac{p_m}{1 + Ae^{-rt}} \tag{2}$$

where $p$ is the population size, $t$ is a time step, and $r$ is the growth rate. $p(t_m) = p_m$ represents the environmental maximum support population size. The logistic model describes the growth of the population as an exponential expression in the form of a sigmoid curve controlled by a carrying capacity due to ecological and resource stresses.

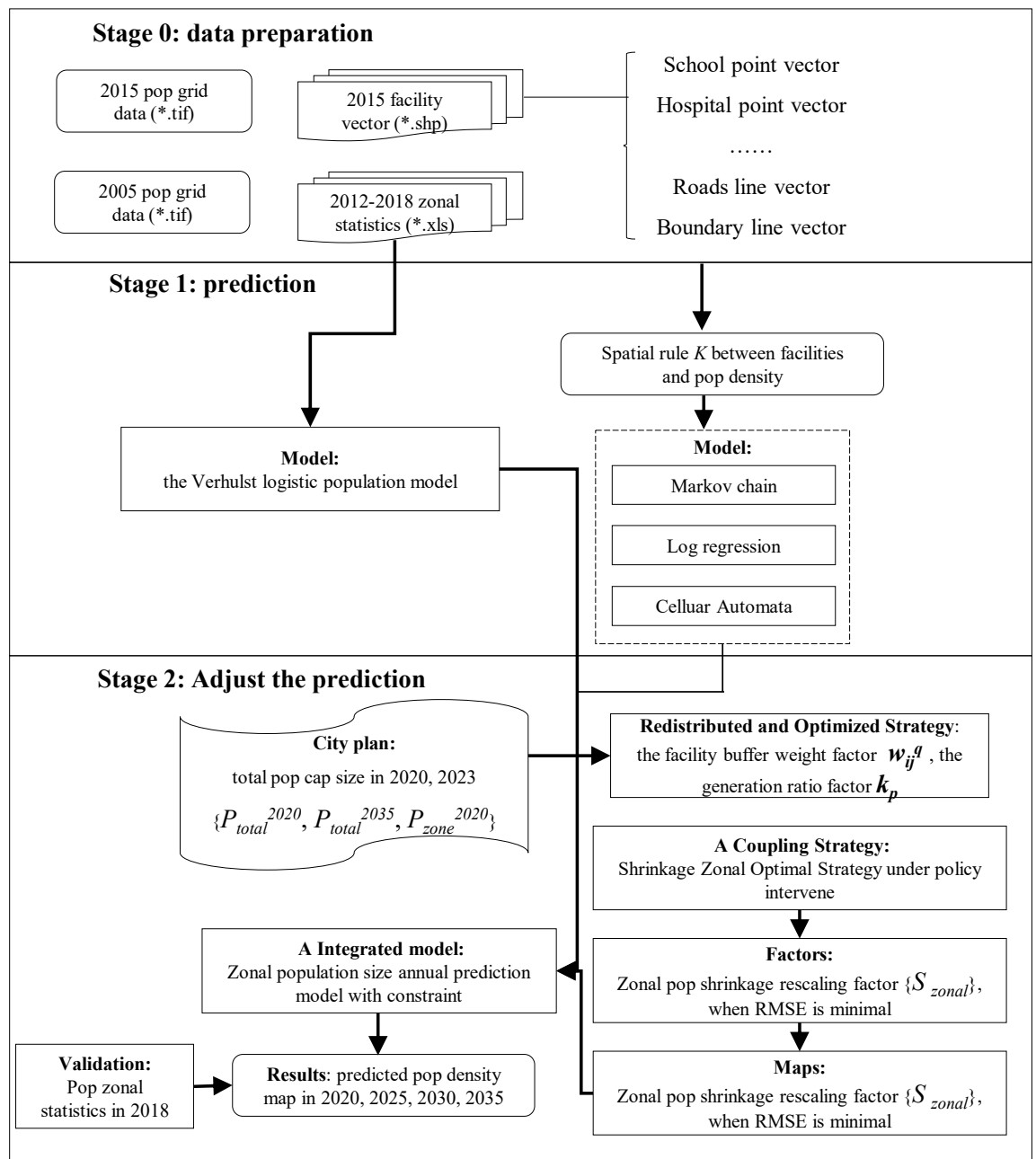

**Figure 3.** Workflow for the study.

The geo-simulation of inhabitants is divided into two parts: the simulation of spatial patterns and the trend of population size in sub-regions. The future sub-regional population dynamics under the "organic decentralization population" policy, as well as the city resource limit is expressed in Equation (3). The first expression in Equation (3) is the population size dynamics prediction based on maximum environmental capability, and the remaining three expressions are the constraint conditions of the "organic decentralization population" policy:

$$
\begin{cases}
\frac{dp}{dt} = r\left(1 - \frac{p}{p_m}\right)dt \\
p_{zone\_six}(T_{2020}) = (1-t)^n * p_{zone\_six}(T_{2014}) \\
\sum_{zone} p_{zone}(T_{2020}) \leq A_1 \\
\sum_{zone} p_{zone}(T_{2035}) \leq A_2
\end{cases}
\tag{3}
$$

where $p(t_0)$, $p(t)$, and $p(t_m)$ are the sub-regional population sizes in years 0 and $t$, and the population cap value with the maximum environmental capability; $t$ and $t_0$ are time, and $r$ is the net population growth rate; parameter $k_m$ (coefficient of the regional population carrying capacity, $k_m = p_m/p_0$) and $r$ has been solved when the RMSE (root-mean-square error between fitting data and original data) is minimized; parameter $t = 15\%$ is the annual decrease rate of population sizes, and parameters $A_1$ and $A_2$ are the cap limit values in 2020 and 2035. By applying MATLAB programming, we could obtain the optimal solution of the equation, representing the annual sub-regional population size.

The spatial distribution of the population was then simulated and spatially optimized according to the needs of different generations. When linking the sub-regional population size $p(t)$ and the regional population distribution $y$, we adopted the minimum error method to calculate the regional adjustment factor $G$. Thus, a new map was obtained with Equation (4):

$$Z_{zone}(t) = \sum_{zone=1}^{16} \sum_{r=0}^{21} y_{zone}^r * G_{zone}^r + \varepsilon \tag{4}$$

where $y_{zone}^r$ is the statistic of the number of pixels in the $zone$th district and the $r$th density level ($zone = 1, \ldots, 16$; $r = 1, \ldots, 21$); $G_{zone}^r$ is the constant in the $zone$th district and the $r$th density level, and $\varepsilon$ is the error.

### 3.2. The Metric of Population Mobility: Barkley Model

Based on Barkley's theory, there are four types of spatial transformation that describe immigration patterns (see Figure 4) [32–34].

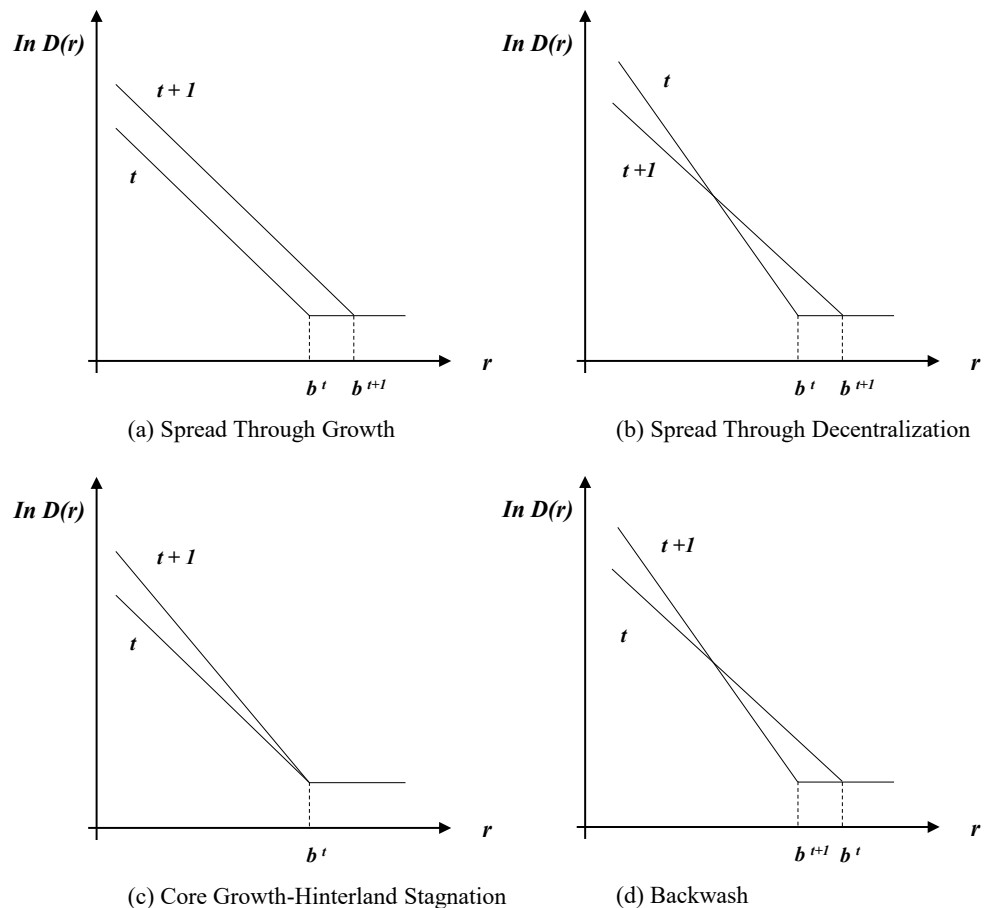

(a) Spread Through Growth

(b) Spread Through Decentralization

(c) Core Growth-Hinterland Stagnation

(d) Backwash

**Figure 4.** Population spatial-density functions and spatial transformation of immigration patterns [34]. Note: $r$ = distance from the center of the urbanized area; $D(r)$ = population density at distance $r$; $b^t$ and $b^{t+1}$ are the points of the threshold value at a greater distance from the nodal center.

### 3.3. Validation of the Model

To verify the model, the predicted map in 2015 was compared with the population density kilometer grid products (Resource and Environment Data Cloud Platform, CHN). Validated by the census data, the population density kilometer grid products were downloaded from the website, classified into five ranges, and distributed over 16 districts. Using zonal statistics, the total number of samples was 5 ranges multiplied by 16 districts. Figure 5 shows the validation of the integrated constrained CA-Markov model by comparing it with the regional demographic statistics at the end of 2018 (from Beijing Demographic Yearbook, 2019). With the 16-region sample data, the correlation value of $R = 0.9995$ and linear equation of $y = 0.99918x + 0.02762$ shows a robust and positive regression between the predicted results and the actual statistics.

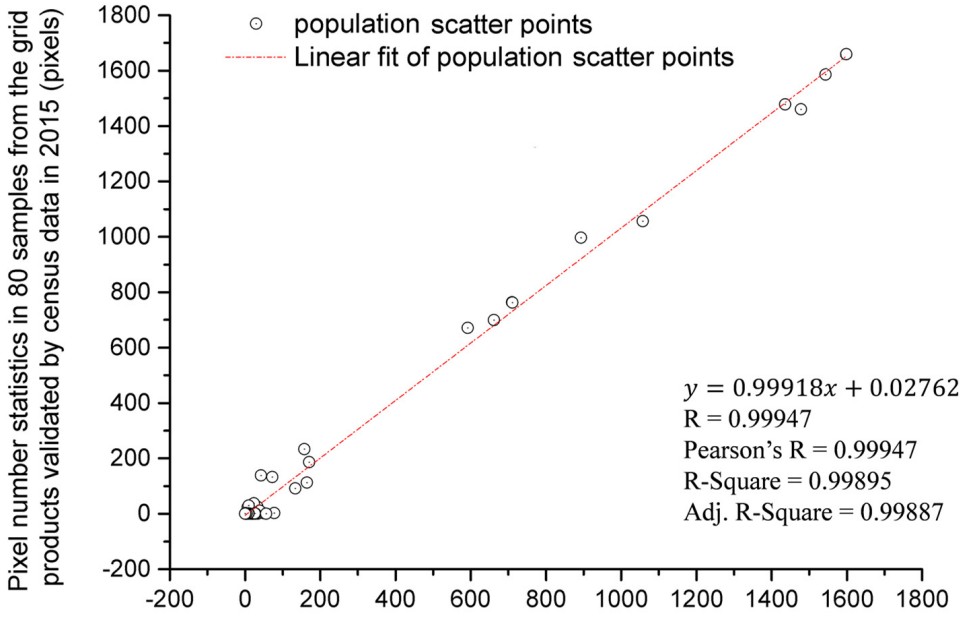

**Figure 5.** Validation of the model when applied to population prediction.

For performance evaluation, two other model results were employed as a comparison. The CA-Markov simulation, CA-Artificial neural network (ANN) simulation, and CA-Decision Tree (DT) simulations were performed using TerrSet 18.21 [35] and a GeoSOS 1.1.1a plugin in ArcGIS [36]. The training parameters of ANN and DT were as follows: diffusion parameter $\alpha = 1$, conversion threshold $\delta = 0.9$, and training accuracy $\kappa = 95.486\%$. The total precision of the simulation was not less than 88.49%, and the kappa coefficient was not less than 0.659. The receiver operating characteristic (ROC) curve and the area under the ROC curve (AUC) values were compared to evaluate the performance of the logistic, ANN, and decision tree models (Figure 6). The final AUC values were 0.909729, 0.862688, and 0.862201 for the CA-Markov, CA-ANN, and CA-DT models, respectively.

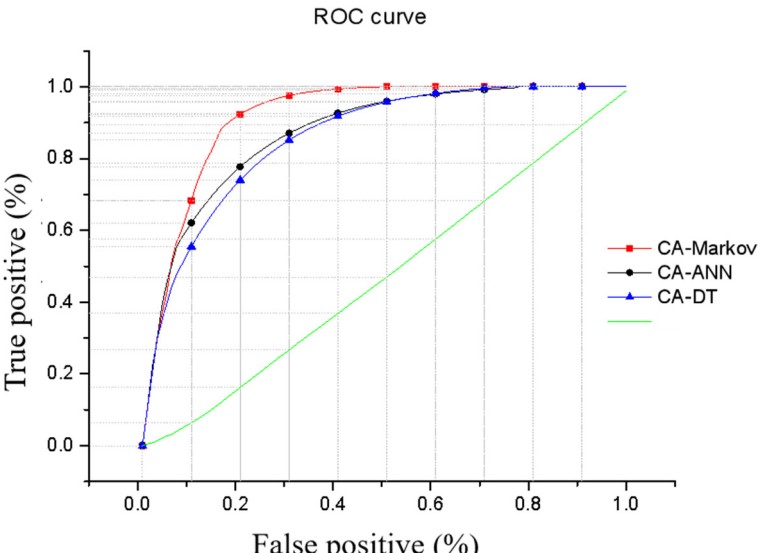

**Figure 6.** ROC curve comparison among the CA-Markov, CA-ANN and CA-DT methods.

## 4. Results

### *4.1. Population Density Geo-Simulation Based on the Integrated Constrained Model in Beijing*

4.1.1. Initial Scenario of Demographic Size Prediction during 2015–2035

By adopting the "isodapanes" model based on Weber's criterion, the static demographic size and density morphology can be described. Tian'anmen Square, Beijing, was the geometric center. In this study, we divided the entire research area into 43 concentric rings with an equal radial gap of $\Delta r = 3$ km. The population size and density variables that fell into each ring were used as the sample data (see Figure 7). After obtaining 43 samples of population size and density in each concentric ring sub-region, we obtained the best fitting curve (see Figure 8). From the results, we concluded that the distribution of the population size was S-shaped.

In Table A1 in Appendix A, 19 single-center models for fitting the population density are compared by the fitting accuracy and number of parameters. The model with the minimum number of parameters and the highest fitting precision was the best choice. The results demonstrated that the Gaussian model had good accuracy, precision, and robustness. By weighing the number of parameters and the fitting precision, the Gaussian-3 function (R-square of 0.98) and the Gaussian-1 function (R-square of 0.99), were selected.

Figure 8a shows the fitting curve of the population size, and Figure 8b describes the population density in concentric rings, both by adopting the Gaussian function. In Figure 8, the 3rd order Gaussian function was proposed for the demographic size fitting results, as it has an accuracy of over 0.98 and three group coefficients. It also demonstrates that the fitting mesh is formed by rotating the fitting line around the *y*-axis.

To evaluate the spatial relationship between facility locations and population density levels, we have two assumptions based on the rule of population data since 2015: (1) the spatial mesh of population density stays the same in various sub-regions. The positions of the facilities (e.g., schools, hospitals, and transportation facilities) are fixed, and the spatial dependence between the population density and the municipal facilities' density remain unchanged for an extended period; and (2) the population growth rate obeyed the Verhulst logistic differential population model during 2014–2018.

Table 2 shows the regression results between 10 types of municipal infrastructure and 20 population density layers during 2005–2015 (sliced by 1000 persons/km², with an upper limit of 24, 000 persons/km² in the densest layer).

# Demographic spatial pattern analysis of Beijing based on multi-ring buffer zone method

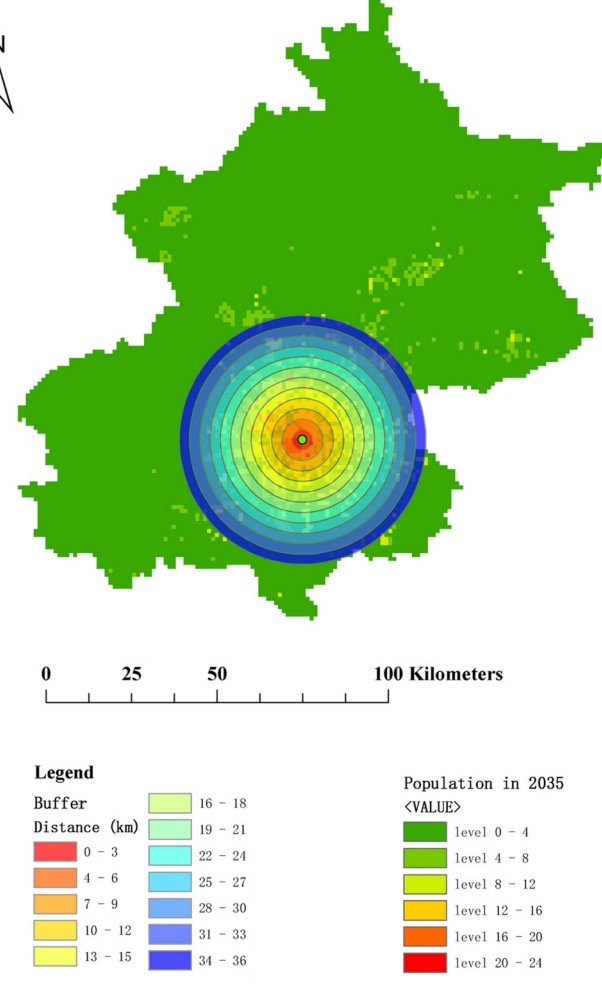

**Figure 7.** The multiple ring buffer analysis method was applied to the analysis of population spatial morphology in Beijing. The span between two adjacent concentric rings was set as Δr = 3 km. There were 43 multiple ring buffers in total, although the above figure shows only 12 rings for illustrative purposes.

**Table 2.** The regression results between the facility variables and population density layer variables.

|  | Restaurant | Roads | Middle School | Primary School | Company | Market | Hospital | Residual Block | Bank | Slope | Intercept |
|---|---|---|---|---|---|---|---|---|---|---|---|
| Coefficient | K1 | K2 | K3 | K4 | K5 | K6 | K7 | K8 | K9 | K10 | B |
| Average value | 6.14 | 5.89 | 5.22 | 4.61 | 2.98 | 2.46 | 2.20 | 1.63 | 1.10 | −3.11 | – |
| General score | 10 | 9 | 8 | 7 | 6 | 5 | 4 | 3 | 2 | 1 | – |

Note: $y_j = \sum k_{ij} * x_{ij} + b_j$, where $\{y_j\}$ expresses the binary maps with different density populations, $x_{ij}$ represents the binary maps with numbers at a certain layer. $K$ is the spatial correlation coefficient, and $b_j$ is the constant.

According to the information in Table 2, the highly correlated factors are "restaurant," "roads," "middle school," "primary school," "company," and "market," while the weakest correlation was obtained from "slope."

For constructing the demographic model after the policy implementation in 2014, the sub-regional population statistics for 2014–2017 were used to train the model parameters $k_m$ and $r$ in the proposed model using MATLAB programming (see Equation (5)), and the values could be solved when the RMSE was minimized. In addition, one year of data (2018) was used to validate the model. The optimum values of $k_m$ and $r$ were determined

by reducing the RMSE between the predicted values and the official statistics. The error minimum of the two equations was RMSE = 2.9551 and RMSE = 2.4450:

$$\begin{cases} p_{zone\_six}(t) = \dfrac{1.05p_0}{1+(1.05p_0/p_0-1)\cdot e^{-(-0.2550t)}}, \\ p_{zone\_ten}(t) = \dfrac{1.25p_0}{1+(1.25p_0/p_0-1)\cdot e^{-0.1876t}} \end{cases} \tag{5}$$

where $P_{zone\_six}$ and $P_{zone\_ten}$ represent the population size of six central districts or ten suburban districts in the $t$th year.

## Sampling data and the Gaussian fitting results

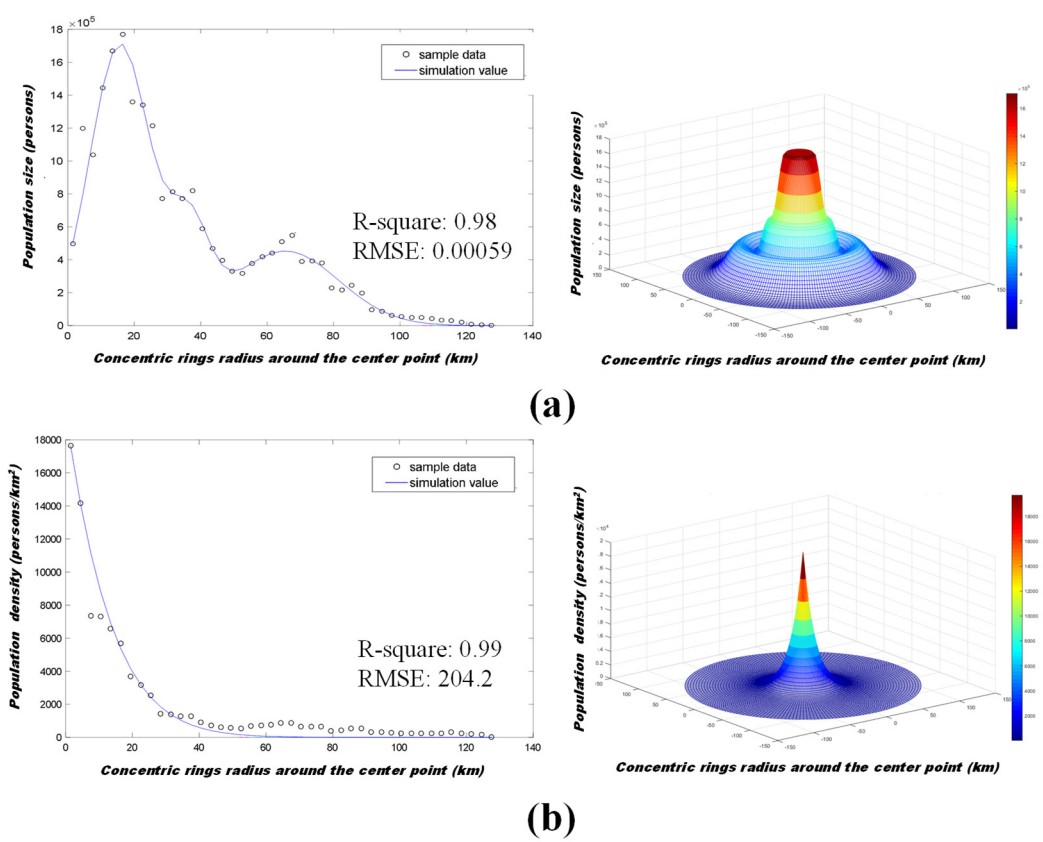

**Figure 8.** A spatial pattern of population size and population density obtained using the expression of the fitting function proposed in this research. Note: Gauss-3 model: $G(r) = a1*exp(-((r - b1)/c1)^2) + a2*exp(-((r - b2)/c2)^2) + a3*exp(-((r - b3)/c3)^2)$; Coefficients (with 95% confidence bounds): $a1 = 5.389 \times 10^5$ ($3.908 \times 10^5$, $6.869^e \times 10^5$); $b1 = 36.56$ (34.48, 38.64); $c1 = 8.38$ (5.299, 11.46); $a2 = 1.71 \times 10^6$ ($1.63 \times 10^6$, $1.79 \times 10^6$); $b2 = 15.74$ (14.89, 16.59); $c2 = 12.83$ (11.45, 14.2); $a3 = 4.533 \times 10^5$ ($3.955 \times 10^5$, $5.112 \times 10^5$); $b3 = 65.39$ (61.66, 69.11); $c3 = 22.86$ (17.4, 28.32); Gauss-1 model: $G(r) = 1.765539 \times 10^4*b*exp(-(r.*r - 2*a*r + a*a)/(2*b*b))$; Coefficients (with 95% confidence bounds): $a = -96.14$ ($-99.94$, $-92.34$); $b = 36.41$ (35.22, 37.61).

The predicted population density grid maps of 2015, 2020, 2025, 2030, and 2035 were obtained based on the integrated constrained CA-Markov model (see Figure 9a).

## Population density distribution simulation map

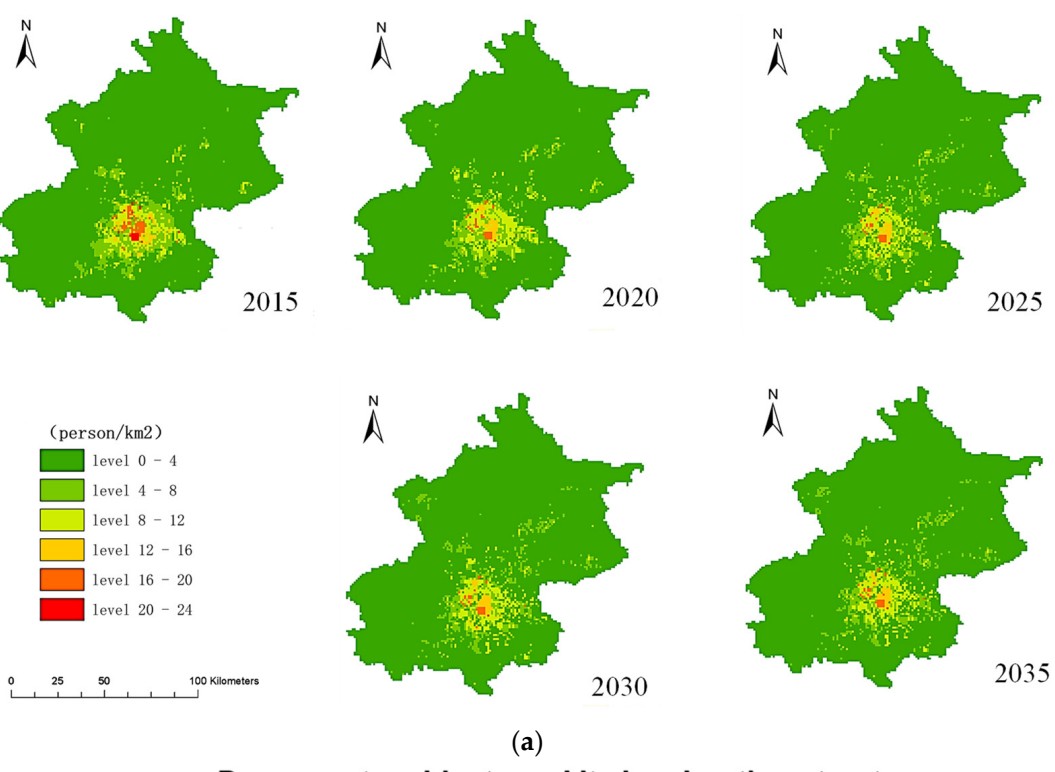

(**a**)

## Permanent residents and its immigrating structure

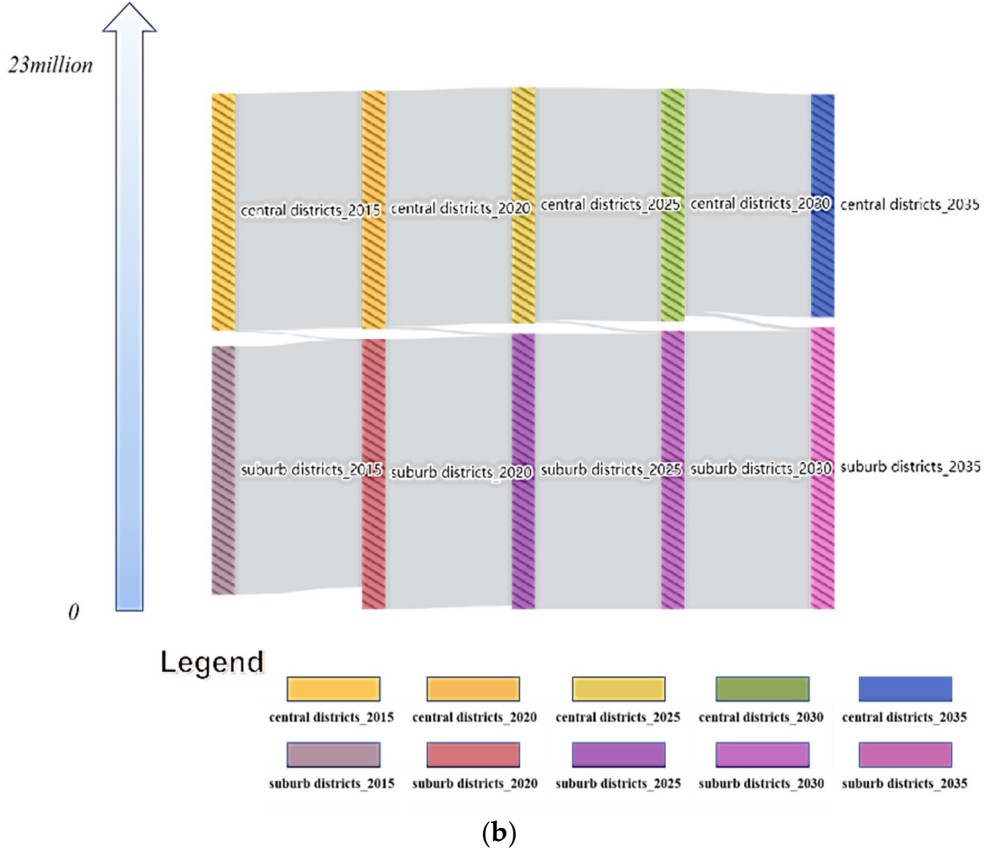

(**b**)

**Figure 9.** *Cont.*

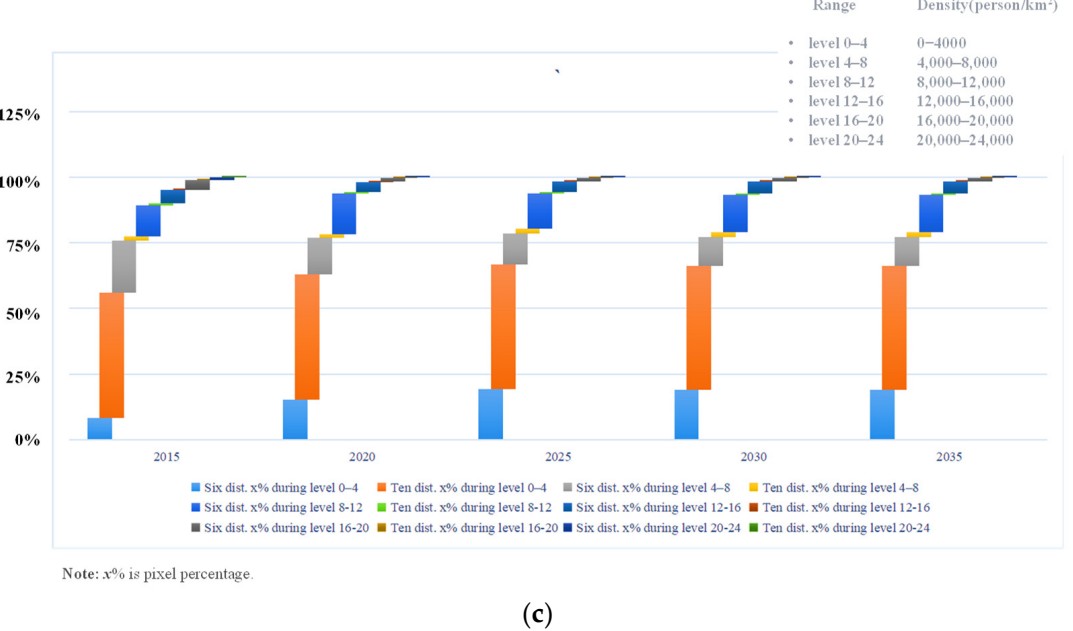

(c)

**Figure 9.** (**a**). Predicted population density level map under policy constraints in 2015, 2020, 2025, 2030, and 2035. (**b**). Permanent inhabitants and the immigrating structure in 2015, 2020, 2025, 2030, and 2035. (**c**). Demographic density stratified statistics under policy constraint in 2015, 2020, 2025, 2030, and 2035.

In Figure 9c, the prediction of permanent resident structure change distributed in urban and suburban areas every five years is another meaningful result. It can be seen from the Sanji chart that the total population size is stable and declining under the trend line of 23 million until 2035. On the other hand, some urban residents migrate to the suburbs or even move out of the city. One main reason for this relates to the municipal government heading out to Tongzhou district, which led to the overall relocation of public institutions, universities, medical facilities, and large enterprises considering lower land costs. Another reason is the urban household registration point system, which has made it difficult for the low-level labor force to settle down.

As Figure 9c shows, due to the outflow of urban residents and immigrants from other cities, the population sizes in the suburban areas increased, and the original suburban areas with low-density populations were occupied by the influx of foreign populations. The results show that the policy played a significant role in controlling regional population dynamics.

4.1.2. Two Scenarios of Population Size Simulation during 2015–2035

Additionally, although the model could predict the population that meets the constraints (Equation (5)), the prediction results were theoretically optimal, but they were not realistic. According to the simulation results, the predicted population size in the hinterland areas was reduced by 80% from 2015 to 2035, which led to a considerable reduction in population and a significant influence on the living conditions of residents. Taking this into account, Scenario II was proposed as a better choice at the cost of the loss of precision. In Scenario II, the parameter $k_m$ was specified as $-0.10$ instead of $-0.255$ during the second stage, 2020–2035. This simulation predicts the population size that satisfies the goal of controlling population while preserving the original civilian living conditions as much as possible. In this case, the plan could be executed smoothly after 2020. Figure 10 illustrates the comparison between the results of the existing model and the new model proposed in this research.

## Prediction results under various scenarios

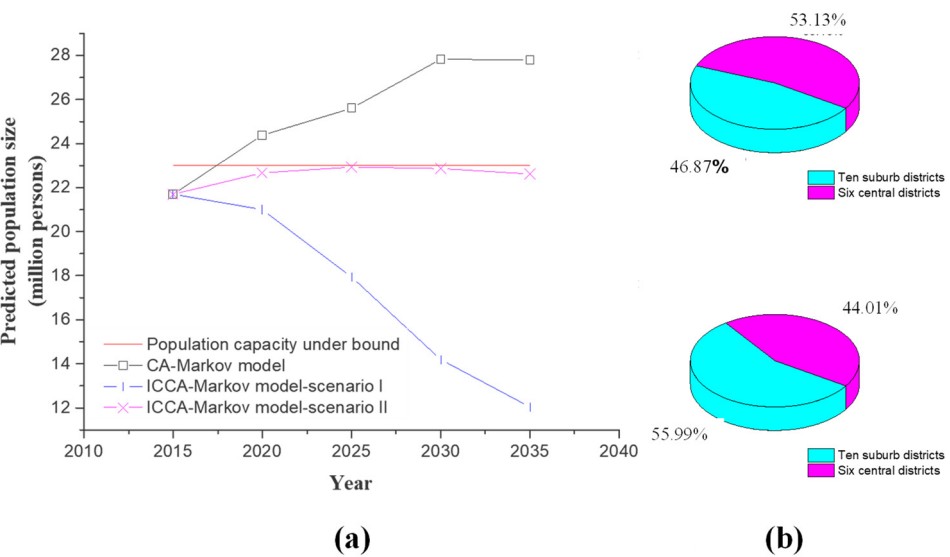

**(a)**                                                       **(b)**

**Figure 10.** (**a**) Result comparison among the traditional CA-Markov model and the integrated constrained CA-Markov model, Scenario I and Scenario II; (**b**) Changes in population structure in two stages, 2015–2020 and 2020–2035 of Scenario II.

### 4.2. Evolutionary Dynamics of Inhabitants Based on Barkley's Theory

Formulating a policy is a vital task, relying on a deep understanding of the mechanism of demographic change adapted to the population control policy in the megacity context. Historically, Tokyo, Keihanshin, and Nagoya did well in the social governance environment, with the development of these megacities being successfully managed [37,38]. With these examples in mind, it is anticipated that the concerned model used in this study will contribute to successful social governance in the Beijing megacity background. The main findings from the model are illustrated in Figure 11, providing vital insights for planning the future city. In Figure 11, the main findings are illustrated as follows: (1) in a certain district (most of the areas), the increase and decrease trend during two time periods are identical. In detail, the overall population size of the six central districts demonstrates negative growth during two stages, 2015–2020 and 2020–2035, while the population size of the ten suburban districts grows or remains unchanged; (2) there is a significant difference across the geographic districts; and (3) the districts with the highest spatial density in the future do not show the fastest growing trends.

From the demographic size and redistribution dynamics, the city transformation stages could be evaluated from the "polarization" development stage to the "orderly" development stage [37–43]. Based on Berkley's theory, immigration patterns could be described by four types of spatial transformation [32–34]. For example, in [44], the population density profiles over time are represented in the form of a negative exponential function to reveal the driving role of policies on population mobility. In Figure 12a, this development mode may be identified as "spread through growth." It may depict an increase in the city center density ($D_0^{t+1} > D_0^t$). In it, little change in the density gradient ($y^{t+l} > y^t$), and an inflection point at a greater distance from the start line ($b^{t+l} > b^t$). Figure 12b shows the "spread through decentralization" mode and represents a stable or declining central city density ($D_0^{t+1} < D_0^t$). In it, a shallower density gradient ($y^{t+l} < y^t$), and a more distant boundary ($b^{t+l} > b^t$). It shows the population spillover primarily toward the metro fringe, referred to as "metropolitan decentralization".

# Demographic size change by districts

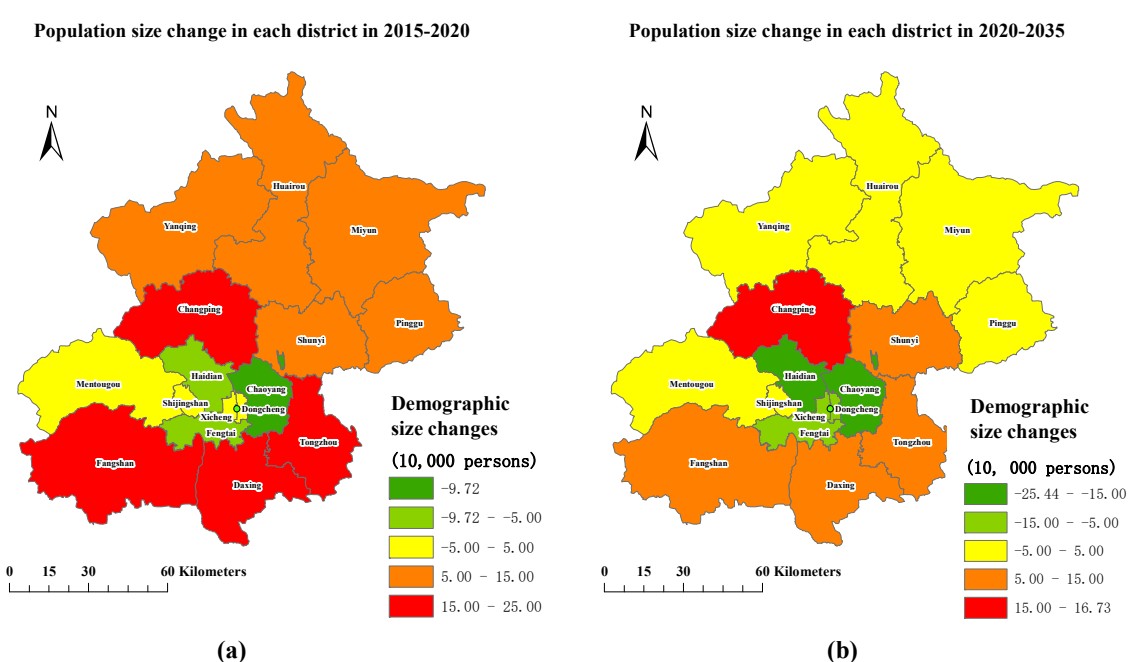

Note: a positive value means increase, a negative one means decrease during a certain stage.

**Figure 11.** Zonal population density change map during two stages, 2015–2020, and 2020–2035.

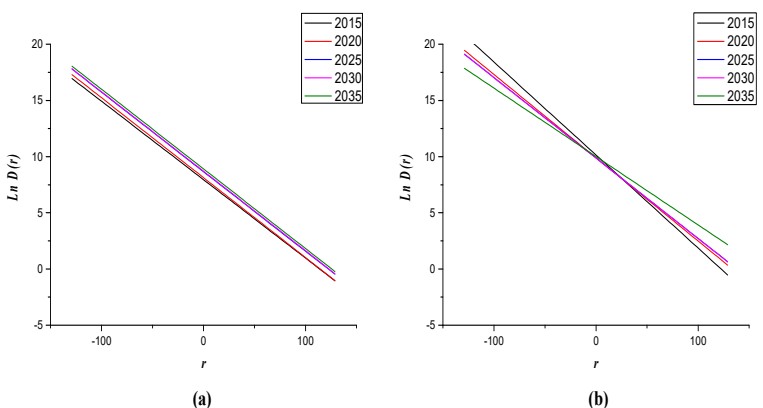

**Figure 12.** Population spatial-density function during two stages, 2015–2020, and 2020–2035. Negative exponential function expression of population density simulation maps in 2015, 2020, 2025, 2030, and 2035. (**a**) The *r-D(r)* functions without constraint, and (**b**) the *r-D(r)* functions under policy constrain.

## 5. Discussion

*Applicability of the Model in Aiding Governmental Planning*

In history, there is no shortage of successful cases of migrant planning. In China, Chengdu Great City—a population dense city located approximately two miles outside Chengdu—attracted 406,211,000 new migrants in the past 10 years [45]. Chengdu main city will gradually become a multi-center and multi-area city. Another example is from Vineyard, Utah, in the U.S., where the population of this technical and industrial satellite town, grew from 139 in 2010 to 15,023 in 2022, making it the 9% growing rate in the country [46]. Over the same period, the populations of 31 cities in the U.S. shrank. In Kenia, Tatu City, the local government (planned for 62,000 residents) will complement and slightly compete with nearby Nairobi, leading to a migration of businesses and homes to

the surrounding suburbia [47]. Despite the history of failed attempts to solve megacity disharmonies through migrant planning, the implementation of the depopulation policy in Beijing may, nonetheless, learn from these failures. For instance, the remodeling of the city of Paris in the 1850s by Baron is a perfect example of a top-down approach. During 1949–1979, eight satellite cities of London attracted a population of 420,000 settlers, but these populations accounted for only 5% of the total population of the region [48]. Another example is from scholars who studied the commuting patterns of migrants in Hong Kong's new towns and found that although the new towns had absorbed a large number of urban residents, the long-term vision proposed in new town plans had not been achieved. Critics of satellite cities point to exorbitant infrastructure development costs and long development timetables, such as Brazil's Fordlandia, England's Harlow, and Egypt's New Cairo [49]. Thus, land-population disharmony remains a severe worldwide problem which has not yet been solved.

## 6. Conclusions and Implications

### 6.1. Contribution, Limitation and Future Study

At present, few studies have conducted relevant simulations and effective evaluations on inhabitant mobility in cities while also considering their urbanization stages. Nevertheless, population planning will have a significant long-term impact on urbanization [50]. This success is manifested in (1) the essence of a city being an agglomeration of population; (2) the contradiction that exists between urban supply and demand, such as environmental pollution, traffic congestion, dense population, housing shortages, and insufficient urban employment opportunities; and (3) the orderly development of the city, which includes an orderly economy, orderly use of space, and orderly population settlement. Among them, the main catalyst for the contradiction is the people. However, it is unclear whether the decentralized population policy guidance will promote the emergence of a polycentric spatial structure within China's big cities.

Similar to many other policy intervention methods, our method is not yet perfect. Firstly, our test results did not consider the human–environment–society relationship agent-based models—especially when there is a discrepancy in the migration pattern among temporary and permanent residents. These task-types are part of a bottom–top model, and policy intervention factors are not sufficiently considered when the survival cost and economic income evaluation are in balance at the same time. Secondly, our integrated CA-Markov method also did not consider the relationship between humans–resources–space in dynamic harmonious status. The CLUE-S model, as one representative of a system dynamics model, assumes that the change of variable in a region is driven by the resources demand and constraint costs in the region, and the internal system factors interact with each other.

However, when prior knowledge is considered to be complete and precisely executed during the decision-making process, our method could perform well and have greater computation cost savings than those of multi-agent models. Besides, when spatial characteristics are considered, our method could reveal causal relationships which can be subsequently used for legislative purposes. Our approach, which has been executed with various scenario simulations, tries to balance accuracy and simulation cost. This integrated constraint CA-Markov approach has been evaluated using official statistics. Furthermore, the ROC curve has been adopted by comparing the performance among the logistic, ANN, and decision tree models, which proved the success and high precision. Our novel approach, which combines regional population policy and temporal population change, results in a more complicated process, but this is an inevitable consequence of providing a more refined solution for urban planning and infrastructure configuration.

For future researches, one of the simulating mechanisms of migrant geographical patterns may be agent-based models. There are several advantages: (a) each agent can have its own attributes and its own states; (b) each agent can be designed to be driven by rules that are its own; (c) each agent can be inserted into a geographical or relational space

that limits its behavior; (d) the behavior of each agent can depend on the behavior of other agents in its local space; (e) each agent possesses variable quantities of information. In the context of a multi-agent system, simulating means asking each agent repeatedly to execute the rules that define them. In the course of these iterations, the aggregated results of agents' behavior can be determined step by step and be reinjected into the behavior of these same agents. Thus, through a dynamic chain of loops connecting different levels of abstraction, agent-based simulation enables the behavior of "low" level entities to be combined to generate the macroscopic regularity that we want to reproduce. The macroscopic policies can be obtained by "low" level entities simulation.

### *6.2. Conclusions*

Our research is beneficial for geographers as an innovative method that couples the existing population spatial patterns with controlling the potential population size; this is also beneficial for urban planners in determining the spatial allocation of municipal facilities. The contributions of this work to the field of urban population planning are summarized as follows:

➢ This research confirmed that a close relationship exists between the inhabitants and various municipal facilities in Beijing. For instance, the highly correlated factors are the density of "restaurants," "roads," "middle schools," "primary schools," "companies," and "hospitals," while the weakest correlation was obtained from the "slope" (see Table 2).

➢ This research proposed a loosely coupled framework between the Verhulst logistic differential population model and the CA-Markov model, which combines the existing population spatial patterns with the potential population size, present pattern, and future trends. The derivation process is presented in Section 3.1, and the synthetic results are presented in Section 4.1.

➢ This research proposed a new Gaussian model for fitting the concentric ring effect of the population density spatial distribution in Beijing, which could be applicable to other cities. By comparing 19 types of distance-density functions, the expression of the proposed function has fewer parameters and higher fitting precision. The expression can be shown as $D(r) = D_0 * b * exp(-(r*r - 2*a*r + a*a)/(2*b*b))$, with two fitting coefficients and an adjusted R-square of 0.9969 (see Table A1 in Appendix A, Figure 8).

➢ By analyzing the evolutionary dynamics of immigrants based on Barkley's theory, this research proved that the development mode under the impact of the depopulation policy intervention might be identified as "spread through decentralization," which reveals the driving role of policies on population mobility (see Figure 12b).

**Author Contributions:** Conceptualization, F.L.; Data curation, W.S.; Funding acquisition, F.L.; Investigation, F.L.; Methodology, W.S.; Project administration, F.L.; Validation, W.S.; Visualization, W.S.; Writing—review & editing, G.P. All authors have read and agreed to the published version of the manuscript.

**Funding:** This research was funded by the National Natural Science Foundation of China: 4160010253, the Basic scientific research business cost project of municipal colleges and universities ZC06: X18092, Natural Science Program of the University's Scientific Research Foundation, the Doctoral Research Initiation Fund: ZF15058.

**Institutional Review Board Statement:** Not applicable.

**Informed Consent Statement:** I have read and I understand the provided information and have had the opportunity to ask questions. I understand that my participation is voluntary and that I am free to withdraw at any time, without giving a reason and without cost. I understand that I will be given a copy of this consent form.

**Data Availability Statement:** Data in this study are available within the article.

**Conflicts of Interest:** The authors declare no conflict of interest.

## Appendix A. Population Density Fitting Models

In Table A1, 19 single-center models fitting the population density are compared by the fitting accuracy and number of parameters. The model with the minimum number of parameters and the highest fitting precision was the best choice. The results demonstrated that the Gaussian model had good accuracy, precision, and robustness. By weighing the number of parameters and the fitting precision, the Gaussian-3 function (R-square of 0.98) and the Gaussian-1 function (R-square of 0.99), were selected.

**Table A1.** Population density fitting models.

| Type of Model | Expression for a General Model | Coefficients | Goodness of Fit | Authors, Year |
|---|---|---|---|---|
| **Linear type** | $D(r) = D_0 + b*r$ with $b < 0$, and $D_0$ is the density at the center | $b = -189.5$ | R-square: $-2.098$ RMSE: 6438 | Commonly used |
| **Polynomial type** | $D(r) = D_0 + b*r + c*r_2$ With $b \neq 0$ and $c < 0$ | $b = -438.3$ $c = -2.55$ | R-square: 0.2501 RMSE: 3206 | Newling, 1971 |
| | $D(r) = a + b*r + c*r_2 + d*r_3$ $a > 0, b < 0, c > 0, d < 0$ | $a = 1.142 \times 10^4$ $b = -449.7$ $c = 5.766$ $d = -0.02337$ | R-square: 0.9739 RMSE: 613.8 | Frankena, 1978 |
| | $D(r) = a*(b + c*(R_0 - r))d$ With $a > 0$ and $R_0$ is radius of the urbanized area | $a = 1.734$ $b = 0.08537$ $c = 0.156$ $d = 2.856$ | R-square: 0.6858 RMSE: 2128 | Mills, 1969 |
| **Power type** | $D(r) = a*rb$ with $a > 0$ and $b < 0$ | $a = 2.682 \times 10^4$ $b = -0.7156$ | R-square: 0.8795 RMSE: 1285 | Smeed, 1963 |
| | $D(r) = a*(Rm - r)b$ With $a > 0, b > 0$ and $Rm$ is the radius of the urbanized area | $a = 2.042 \times 10^{-11}$ (*fixed at bound*) $b = 7.04$ | R-square: 0.994 RMSE: 287.3 | Commonly used |
| **Exponential type** | $D(r) = D_0*exp(b*r)$ with $b < 0$ | $b = -0.07619$ | R-square: 0.9992 RMSE: 102.3 | Clark, 1951 |
| | $D(r) = D_0*exp(b*r*r)$ with $b < 0$ | $b = -0.005431$ | R-square: 0.9973 RMSE: 191.2 | Tanner, 1961 |
| | $D(r) = D_0*exp(b*r)*rc$ With $b < 0$ and $c > 0$ | $b = -0.07619$ $c = 3.64 \times 10^{-14}$ (*fixed at bound*) | R-square: 0.9971 RMSE: 198.8 | Aynvarg, 1969 |
| | $D(r) = D_0*exp(b*sqrt(r))$ With $b < 0$ | $b = -0.385$ | R-square: 0.9964 RMSE: 218 | Commonly used |
| | $D(r) = D_0*br$ with $b > 0$ | $b = 0.9266$ | R-square: 0.9992 RMSE: 102.3 | Commonly used |
| | $D(r) = D_0*exp(b*r_2 + c*r)$ with $b > 0$ and $c > 0$ | No fitting model. | NA | Newling, 1969 |
| | $D(r) = a*exp((b*r + c*r_2)*r\char`^d)$ $D(r) = a*exp((c*r_2)*rd)$ $D(r) = a*exp((b*r + c*r_2 + d*r_3)*r_e)$ with $a > 0, b > 0, c < 0, e < 0$ | No fitting model. | NA | Zielinski, 1979 |
| | $D(r) = D_0*exp(b*r + c/r)$ With $b < 0$ and $c > 0$ | $b = -0.07635$ $c = 0.1718$ | R-square: 0.8977 RMSE: 377.6 | McDonald and Bowman, 1976 |
| **Logarithm type** | $D(r) = a + b*log(r)$ With $a > 0$ and $b < 0$ | $a = 1.596 \times 10^4$ $b = -3608$ | R-square: 0.8084 RMSE: 2162 | Commonly used |

**Table A1.** *Cont.*

| Type of Model | Expression for a General Model | Coefficients | Goodness of Fit | Authors, Year |
|---|---|---|---|---|
| **Fourier type** | $D(r) = a + b*cos(r*w) + c*sin(r*w)$ $Fourier_1$, *the number of items equals to one.* | a0 = $4.014 \times 10^{11}$ a1 = $-4.014 \times 10^{11}$ b1 = $9.588 \times 10^{7}$ w = $-2.82 \times 10^{6}$ | R-square: 0.7399 RMSE: 1936 | Commonly used |
| **Gaussian type** | $D(r) = D_0*b*exp(-(r*r - 2*a*r + a*a)/(2*b*b))$ | a = $-96.14$ b = $36.41$ | R-square: 0.997 RMSE: 204.2 | Liu, 2019 |
| | $D(r) = a_1*exp(-((r - b_1)/c_1)2) + a_2*exp(-((r - b_2)/c_2)2)$ | a1 = $6.076 \times 10^{5}$ b1 = $-79.14$ c1 = $40.26$ a2 = $2.348 \times 10^{19}$ b2 = $-1609$ c2 = $268.4$ | R-square: 0.9803 SE: 643.5 | Commonly used |
| | $D(r) = a_1*exp(-((r - b_1)/c_1)\hat{}2) + a_2*exp(-((r - b_2)/c_2)\hat{}2) + a_3*exp(-((r - b_3)/c_3)\hat{}2)$ | a1 = $1.482 \times 10^{4}$ b1 = $1.146$ c1 = $4.955$ a2 = $4455$ b2 = $12.97$ c2 = $8.034$ a3 = $2.228 \times 10^{17}$ b3 = $-2723$ c3 = $481.8$ | R-square: 0.9959 RMSE: 306.7 | Commonly used |

**Note:** (1) The robust method for the evaluation of all the fitting functions is the LAR method. (2) The fitting method is the non-linear least squares method. (3) If the fitting equation could not be computed, "No fitting model" is shown.

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
