# Peer review of "Demographic Spatialization Simulation under the Active “Organic Decentralization Population” Policy"

_sustainability, doi:10.3390/su142013592_

Round 1

Reviewer 1 Report

Dear authors,
First of all your topic regarding urbanization is an important one, considering that there are many places across the world that require proper planning and organization for population management. You focusing on the Beijing-Tianjin-Hebei agglomeration makes sense considering the contribution of industrialization to population, which makes your research significant.

Reading your article, I found one typo:

Line 114. The sentence starts with ".e" I believe it should be ". The"

Beyond that, your citations are good as you describe the methods. However, I feel that you could cite more rigorously in your introduction portion. When I see a sentence like the following:

Recently, an increasing number of geographers and urban planners have been promoting the decentralization of population by reducing the multi-functionality of cities, guiding house prices, or influencing enterprise relocation. 

with no citations [x, y, z], at the full stop, then it is hard to figure who are these geographers and in which study they promoted this. Such citations would be better to guide the readers as well. Your introduction part must be checked throughout for proper citations.

I have no problems with your methodology as you described the methods you utilized properly, showing results and providing valuable discussion.

About the human-environment-society relationship agent-based models you mentioned in Section 5.2., if this was included in your research, do you think it would make a significant contribution? If so, you could also indicate this as the future direction of your research and perhaps provide a few insights there. 

Good luck!

Author Response

Thank you for your review of our paper. We have answered each of your points below.

Point 1: Line 114. The sentence starts with ".e" I believe it should be ". The"

Response 1: Thank you for your alert. I will look through the whole manuscript again to prevent such a clerical error.

point 2: Your citations are good as you describe the methods. However, I feel that you could cite more rigorously in your introduction portion. with no citations [x, y, z], at the full stop, then it is hard to figure who are these geographers and in which study they promoted this. Such citations would be better to guide the readers as well. Your introduction part must be checked throughout for proper citations.

Response 2: In order to indicate where specific information in an essay or journal came from, we write immediately the borrowed information with an in-text citation. If there are different citations corresponding to different words which are of coordination relation, the citations' position are right after each word, but not at the full stop. I will look through the whole manuscript again to prevent such a phenomenon.

Point 3: About the human-environment-society relationship agent-based models you mentioned in Section 5.2., if this was included in your research, do you think it would make a significant contribution? If so, you could also indicate this as the future direction of your research and perhaps provide a few insights there.  

Response 3: The agent-based models have not been studied in this manuscript, although it is a potential simulation model for discussing spatial migration mechanism. On the basis of rules of behaviour and interaction, it aims to imitate the details of mechanism or actual process.

For simulating the mechanism of migrant geographical pattern, there are several advantages.

  1. a) Each agent can have its own attributes and its own states; b) each agent can be designed as driven by rules that are its own; c) each agent can be inserted into a geographical or relational space that limits its behaviour; d) the behaviour of each agent can depend on the behaviour of other agents in its local space; e) each agent possesses variable quantities of information (Epstein, 2006). In the context of a multi-agent system, simulating means asking each agent repeatedly to execute the rules that define them. In the course of these iterations, the aggregated results of agents’ behaviour can be determined step by step and be reinjected into the behaviour of these same agents. Thus, through a dynamic chain of loops connecting different levels of abstraction, agent-based simulation enables the behaviour of “low” level entities to be combined to generate the macroscopic regularity that we want to reproduce.The macroscopic policies can be obtained by simulationof “low” level entities behavior guided by government interventions and personal incentives.

Reviewer 2 Report

Thank you for the invitation to review the manuscript entitled: “Impact of active “organic decentralization population” policy on future demographic spatialization simulation”. The “decentralize and population cap” policy implemented in Beijing, China, was investigated in the paper. The paper's title is informative, and the references are also relevant. The sections of the manuscript were well-organized, but I detected some gaps. In my opinion, the manuscript can be published in the journal after revisions.

My remarks are:

1.     The authors should give brief information about the decentralize and population cap policy. 

2.     The B-A-B-C mode mentioned in the Introduction should be explained.

3.     The used data and Methodology were not introduced in the Introduction. 

4.     The readability and visibility of figures are low. Therefore, their resolutions of them should be improved. Furthermore, the figures should be cited in the text (for ex., Figure 9b).

5.     The CA-ANN, CA-DT, and CA-ANN models should be described in the Methodology.

6.     The conclusions are inadequate, so they should be improved according to research questions and results.

Author Response

Thank you for your review of our paper. We have answered each of your points below.

Point 1: The authors should give brief information about the decentralize and population cap policy. 

Response 1: In this research, the citation of "organic decentralization project" is originally proposed in "Beijing General Urban Planning (2016-2035)", cited in [6]. To flatten the spatial density curve in the central six districts, prevent the further fanatical migrants influx from the suburbs and other provinces, and promote the land urbanization in the suburbs, this population cap policy has been proposed through the in-depth investigation. Other studies, such as those posed in [7], a good mix of (fiscal and regulatory) interventions have achieved effective performance in Delhi.

point 2: The B-A-B-C mode mentioned in the Introduction should be explained.

Response 2: Here, the meaning of " B-A-B-C mode" is the spiral growth of a creative city, from polarization to transitional stages, then from transitional to orderly stages, and so on. Figure 1 illustrates this meaning.

point 3: The used data and Methodology were not introduced in the Introduction.

Response 3: The modelling data comes from the statistical yearbooks and RS (Remote Sensing) image interpretation products, while validated by the district-level street statistics data. For example, this spatial population kilometer grid data (2005,2010) is from Global Change Research Data Publishing & Depository, CHN, which is used for modelling.

In this research, an integrated constrained Verhulst and CA-Markov model was proposed.

point 4: The readability and visibility of figures are low. Therefore, their resolutions of them should be improved. Furthermore, the figures should be cited in the text (for ex., Figure 9b).

Response 4: These figures will reach 300 ppi as much as possible, and the font size could be readable at 100% zoom ratio. Of course, the figures will be cited in the text.

point 5: The CA-ANN, CA-DT, and CA-Markov models should be described in the Methodology.

Response 5: To obtain reliability of results and performance evaluation, the existing models have been used for comparison, such as CA-ANN (Cellular Automata-Artificial Neural Network), CA-DT (Decision Tree), and CA-Markov models. Considering that these methods are common and the size of the article has reached the upper limit, these three methods are not suitable for detailed introduction in the Methodology. However, they could be cited in the text for tracking the details.

point 6: The conclusions are inadequate, so they should be improved according to research questions and results.

Response 6: The revised Conclusion part consists of three aspects: original conclusion, contributions, and prospect. The model was tested on planning-oriented creative Beijing city, and was proved to be a successful application. It is an easy-to-use and valuable solution for predicting the immigrant spatial pattern, for evaluating the balance relationship with land resources. The future developments of this work will involve the improvement of the performance of the models and, at the same time, expand it, considering the most promising technologies, to the human-environment-society relationship agent-based models.

Round 2

Reviewer 2 Report

The authors have considered the suggested comments, but Fig. 9b still was not cited in the text.

Author Response

Thank you for your alert. I will look through the whole manuscript again to prevent such an error. A new paragraph like this has been added:

In Fig. 9(c), the prediction of permanent resident structure change distributed in urban and suburban areas every five years is another meaningful result. It can be seen from the Sanji chart, the total population size is stable and declining under the trend line of 23 million untill 2035. on the other hand, some urban residents migrate to the suburbs or even move out from the city. One main reason for that is affairs of the municipal government heading out to Tongzhou district, which led to the overall relocation of public institutions, universities, medical facilities and large enterprises considering lower land costs. Another reason is the urban household registration point system, which has made it difficult for low-level labor force to settle down.

I have seen the suggestions in the Review Report Form, and they have high generality and it made me confused to modify them. If you could point it out the lines, I will correct it carefully. As for the research design, questions, hypotheses and methods, I particularly hope to improve. Thank you again for taking the time to give me valuable suggestions.